



# Impact of assimilating NOAA VIIRS Aerosol Optical Depth (AOD) observations on global AOD analysis from the Copernicus Atmosphere Monitoring Service (CAMS)

Sebastien Garrigues[1], Melanie Ades[1], Samuel Remy[2], Johannes Flemming[1] , Zak Kipling[1], Istvan laszlo[3] , Mark Parrington[1], Antje Inness[1], Roberto Ribas[1], Luke Jones[1], Richard Engelen[1],Vincent-Henri Peuch[1]

1: ECMWF, Reading, RG2 9AX, UK
2: HYGEOS, Lille, France
3: Center for Satellite Applications and Research, NOAA/NESDIS,
College Park, USA

*Correspondence to*: Sebastien Garrigues (sebastien.garrigues@ecmwf.int)



**Abstract**. Global monitoring of aerosols is required to analyse the impacts of aerosols on air quality and to understand their role in modulating the climate variability. The Copernicus Atmosphere Monitoring Service (CAMS) provides near real time

forecasts and reanalyses of aerosols using the ECMWF Integrated Forecasting System (IFS), constrained by the assimilation of MODIS and PMAp Aerosol Optical Depth (AOD). Given the potential end-of-lifetime of MODIS AOD, implementing new AOD observations in the CAMS operational suite is a priority to ensure the continuity of the CAMS forecast performances. The objective of this work is to test the assimilation of the NOAA VIIRS AOD product from S-NPP and NOAA20 satellites in the IFS model. Simulation experiments assimilating VIIRS on top or in place of MODIS were carried out from June 2021

to November 2021 to evaluate the impacts on the AOD analysis.

For maritime aerosol background, the assimilation of VIIRS and the use of VIIRS from NOAA20 as an anchor reduce the analysis AOD values compared to MODIS-based experiments, in which the analysis values were too high due to the positive bias of MODIS/TERRA over ocean. Over land, the assimilation of VIIRS induces a large increase in the analysis over biomass burning regions where VIIRS shows larger AOD than MODIS. For dust source regions, the analysis is reduced when VIIRS

is assimilated on top of or in place of MODIS, particularly over the Sahara, Arabian Peninsula and few places in Asia in the July-August period. The assimilation of VIIRS leads to an overall reduction of the bias in AOD analysis evaluated against AERONET measurements with the largest bias reduction over Europe, desert, and maritime sites.



## 1 Introduction

Satellite observations of aerosol optical depth (AOD), which represents the total atmospheric column mass extinction, offer a
great potential to resolve the spatial and temporal variations of aerosols at both regional and global scales (Kokhanovsky et al,
2007). They have frequently been used to reduce the impacts of uncertainties in aerosol model initial conditions on the aerosol
forecast through data assimilation techniques. Data assimilation (DA) consists in producing an optimal estimate of the
atmospheric state (denoted hereafter the analysis), by combining the observation and the model background information in an
optimal way which explicitly accounts for both the observation and model background errors (Courtier et al., 1994). Generoso
et al. (2007) showed that the assimilation of coarse and fine mode AOD retrieved from POLDER in the LMDz-INCA model
improves the aerosol burden estimation over the artic region. Besides, MODIS AOD from TERRA and AQUA satellites, which
have been frequently used in global aerosol data assimilation systems, was proved to improve the forecast performances
compared to free running experiments (Benedetti et al, 2009; Zhang et al., 2008, 2014; Sic, et al., 2016; Di Tomaso et al,
2017). Xian et al., 2019 showed that the spread between aerosol models is reduced when they are constrained by DA compared
to free running aerosol models. Forecast improvement particularly concerns biomass burning and dust outbreak events along
with regions associated with large emission uncertainties such as polluted hot spots in Southeast Asia (Sic et al., 2016; Choi
et al., 2020; Escribano. et al., 2022). Aerosol forecast is generally improved up to 72h lead time after which the impact of the
uncertainties in the emission data set is larger than that of initial conditions. Aerosol data assimilation systems are based on
distinct data assimilation techniques which includes variational (Benedetti et al., 2009), Kalman filter (Schutgens et al., 2010)
and ensemble (Rubin et al., 2016,2017, Di Tomaso et al., 2017) methods. Most data assimilation systems assimilate AOD,
which gives information on the total atmospheric column aerosol mass extinction. They rely on the model background to
redistribute the analysis increments on the different species simulated by the model according to their fractional contribution.
Other systems assimilate species-dependent observations (Di Tomaso et al., 2017). For example, the dust forecast from the
Met Office Unified Model relies on the assimilation of MODIS AOD which has been filtered to assimilate only dust
observations (Pope et al., 2016). More recently, Escribano. et al. (2022) have shown the benefit of the joint assimilation of
spaceborne dust extinction coefficient profile and dust optical depth observations. While most data assimilation systems have
been exploiting single AOD products, the current challenge lies in designing efficient multi-satellite AOD assimilation
schemes to increase the resilience of the data assimilation system against instrument failure and to enhance the accuracy of the
analysis by maximizing the spatial and temporal coverage of the observations and exploiting enhanced information content
from new generation of satellite retrievals (Benedetti et al., 2018).

The Copernicus Atmosphere Monitoring Service (CAMS) produces global forecasts and reanalyses of aerosol, trace gas
species and greenhouse gases (GHG) by using the Integrated Forecasting System (IFS) developed at ECMWF and a 4D-VAR
data assimilation scheme (Flemming et al., 2015; Inness et al., 2019; Huijnen et al., 2019, Remy et al., 2019). For aerosol,
MODIS AOD from TERRA and AQUA satellites produced by NASA (Levy et al., 2013) and the Polar Multi-Sensor Aerosol
Optical Properties (PMAp) AOD produced by EUMETSAT from instruments on board MetOp-B, and MetOp-C



(EUMETSAT, 2021) are assimilated in the IFS. However, both Terra and Aqua satellites are now drifting, meaning their respective mean local time of observation is changing. Besides, the satellite altitude is slowly decreasing with time, leading to decrease in pixel size on the ground and increasing gaps between orbits which may negatively impact the aerosol analysis and the performances of the forecast. In this context, VIIRS AOD has been recently compared with MODIS, PMAp and the

modelled AOD to prepare for its future assimilation in the IFS and increase the resilience of the DA system against MODIS instrument failure or product disruption (Garrigues et al., 2022). VIIRS AODs have been produced by both NASA (Sayers et al., 2018, Hsu et al., 2019, Sawyer et al., 2020) and NOAA (Laszlo and Liu, 2022). At the time of its implementation in CAMS, only the NOAA product was produced in near real time (NRT) and provided retrievals from both S-NPP and NOAA20 platforms. Garrigues et al., 2022 have evaluated the consistency of the NOAA VIIRS (v2.r1) AOD with MODIS (C6.1) AOD

at the model spatial resolution over both land and ocean from December 2019 to May 2021. They reported that VIIRS AOD is generally lower than MODIS over ocean and larger over biomass burning and some dust source land regions. Besides, VIIRS also shows larger spatial coverage than MODIS related to its higher spatial resolution and larger swath. Finally, the study indicates significant offsets between retrievals from the same instrument on-board distinct satellite platforms: TERRA/MODIS and SNPP/VIIRS are frequently larger than AQUA/MODIS and NOAA20/VIIRS respectively. This is related to biases in

input radiances which translate directly to the AOD retrieval. These biases between the same instrument but on-board different platforms need to be accounted for when the observations are used in the assimilation process. Ideally, well-characterized ground-based observations with very small bias are used to 'anchor' the DA system, but lacking those the satellite dataset that is considered to be the most reliable can be used without bias correction as an 'anchor' against which the other observations are bias corrected.

The objective of this paper is to evaluate the impact of assimilating the NOAA VIIRS AOD product from S-NPP and NOAA20 satellites on the AOD analysis from the IFS. Distinct experiments assimilating VIIRS on top or in place of MODIS were conducted from June 2020 to November 2020. This period includes both typical seasonal variations of aerosol load as well as exceptional aerosol outbreaks related to dust storms and extreme biomass burning events (Fig. 1).

Section 2 provides a description of the satellite AOD observations used in this work. Section 3 presents the IFS model, the
4D-VAR data assimilation method used in IFS and the simulation experiments designed for this work. The results are summarized in Sect. 4. and are discussed in Sect. 5. Conclusion and recommendations from this work are given in Sect. 6.

## 2 Satellite AOD products

The similarities and differences between the NOAA VIIRS and NASA MODIS AOD products are summarized here. More detailed information on MODIS and VIIRS algorithms, their relative performances assessed at the IFS spatial resolution as
well as the PMAp product which is assimilated over ocean but is not the focus of this paper, can be found in Garrigues et al., 2022.



MODIS and VIIRS are two imaging radiometers. VIIRS has a larger swath than MODIS. Besides, it shows reduced pixel deformation at the edge of the swath which should limit geometry-induced retrieval biases. MODIS AOD product has a ~10 km spatial resolution while VIIRS is retrieved at the spatial resolution of 0.750 km. However, the VIIRS product used in CAMS is the BUFR version for which data was thinned by using every second column from every second row of the original NETCDF product distributed by NOAA, resulting in an effective resolution of 1.5 km.

MODIS product includes two separate retrieval algorithms: the dark target (DT, Levy et al., 2013) over land and ocean and the deep blue (DB, Hsu et al., 2019) over land only. MODIS and VIIRS algorithms exploit similar spectral information content from selected visible (VIS) and shortwave infrared (SWIR) bands. They rely on a similar ocean surface reflectance model which represents the contributions from sun-glint, underwater and whitecap. Over vegetated land surfaces, MODIS (DT and DB) and VIIRS exploit a similar spectral constraint approach (Kaufman et al., 1997) to estimate the surface reflectance in the visible domain (Levy et al., 2013., Hsu et al., 2013, Laszlo and Liu, 2020). Over bright and complex surfaces, both MODIS DB and the VIIRS algorithms exploit a surface reflectance database and account for the surface anisotropy. VIIRS and MODIS DT use similar aerosol models over ocean adapted from Remer et al., 2005. The aerosol models used over land in the NOAA VIIRS AOD algorithm are the same as those in the MODIS DT algorithm; they were adopted from MODIS collection 5 AOD algorithm as described in Levy et al., 2007. The difference is that the MODIS DT algorithm prescribes the aerosol models geographically (allowing only for dust contribution dynamically) while the NOAA VIIRS algorithm selects one of the candidate models for each pixel by the evaluation how well reflectances calculated for each model match the reflectances observed in selected spectral bands. All products are associated with quality assessment information and provide pixel-level uncertainty information. For MODIS DT the uncertainty is computed as a function of AERONET AOD while for the DB and VIIRS it is a prognostic function of the retrieved AOD.

## 3 Model and data assimilation

### 3.1 Model

The experiments were performed using IFS cycle 47R3 which was run at a horizontal spectral resolution of TL511 (equivalent to a grid size of about 40 km) and comprises 137 atmospheric levels (0.01 to 1013 hPa). IFS exploits a semi-Lagrangian scheme (Temperton et al. 2001) and a mass fixer (Diamantakis and Flemming, 2014) to simulate the transport of atmospheric tracers. Trace gas species and chemistry processes are simulated using the modified Carbon Bond 05 (CB05) chemistry scheme (Flemming et al., 2015 and Huijnen et al.,2019). A bulk-bin scheme is used for aerosol modelling (Remy et al., 2022). 14 aerosol tracers are simulated which include 7 distinct species (dust, sea salt, organic matter, black carbon, sulfate, nitrate and ammonium), 3 bin sizes for dust and sea salt, fine and coarse mode for nitrate and hydrophilic and hydrophobic for black carbon and organic matter. Emission of sea-salt and dust are simulated using the IFS meteorological variables. Biomass burning emission estimates are provided by the Global Fire Assimilation System (GFASv1.4) which relies on the assimilation of MODIS Fire Radiative Power observations of (FRP) (Kaiser et al., 2012). The CAMS-GLOB-





ANT 4.2 (Elguindi et al., 2020) and the CAMS-GLOB-BIO v1.1 (Sindelarova et al., 2014) emission inventories provide the

rest of the static primary aerosol for anthropogenic sources and biogenic sources, respectively. More detailed on the aerosol

model implemented in IFS can be found in Remy et al. (2019, 2022).

### 3.2 Data assimilation method

An incremental 4D-VAR assimilation scheme is applied to both meteorological and atmospheric composition variables using

a 12-h assimilation window (Courtier et al., 1994). In CAMS the 12 h assimilation windows are defined from 03:00 to 15:00

(denoted 12z) and from 15:00 to 03:00 (denoted 00z) UTC. 4D-VAR works by minimising a cost function which has as input

a control vector based on the initial conditions. The cost function measures the differences between the model's background

fields, forecast from the initial conditions, and the observations throughout the assimilation window and is weighted by the

error statistics of both the background and the observations. The incremental form of 4D-VAR used in the IFS means that a

sequence of minimisations is carried out on a lower resolution linearisation of the full non-linear model. In the CAMS

operational system, these so-called inner loop minimisations are currently carried out at TL95 and TL159, which correspond

to horizontal resolutions of 210km and 125km respectively.

The variable used in the control vector for aerosols is a total aerosol mass mixing ratio, defined as the sum of the aerosol

species, since there is not enough information in the observations of AOD to constrain all 14 aerosol bins. The total aerosol

analysis increments produced by the assimilation are repartitioned into the individual components according to their fractional

contribution to the total aerosol mass (Benedetti et al., 2009).

The background errors used in the cost function for the total aerosol control variable are calculated with the National

Meteorological Centre (NMC) method (Parrish and Derber, 1992), using differences between pairs of background fields which

have the statistical characteristics of the background errors. The background error covariance matrix is given in a wavelet

formulation (Fisher 2004, 2006). This allows both spatial and spectral variations of the horizontal and vertical background

error covariances. The CAMS background errors are constant in time.

The observation operator used to transform the aerosol mass mixing ratios into AOD consists of a Look Up Table (LUT) of

aerosol optical properties which were computed assuming externally mixed aerosol species, spherical shape and a lognormal

size distribution prescribed for each aerosol species. The relative humidity derived from the IFS meteorological variables is

used to represent its impact on the optical properties of hygroscopic aerosol. The aerosol mass of each aerosol species is first

interpolated at the location and time of the observation and then integrated vertically. AOD at 0.55 μm for each individual

aerosol species is computed from the extinction coefficient stored in the LUT and then the total AOD is calculated as the sum

of the individual species AOD.

Once an initial condition has been optimized by the 4D-VAR minimisation process, a five-day aerosol forecast is run from

00z for the 15.00 to 03.00 assimilation window and from 12z for the 03.00 to 15.00 assimilation window.




### 3.3 Assimilation of MODIS and PMAp in IFS

In the CAMS operational suite, MODIS Collection 6.1 is assimilated over ocean and land. DT is preferentially used, and DB

is employed to gap-filled DT over land. PMAp version 2.1 is assimilated over ocean only. In this work, the assimilation of the NOAA VIIRS AOD is tested over both land and ocean.

Best quality retrievals (according to QA information) are selected for the assimilation. Thinning and superobbing techniques are applied to reduce the number of observations, to ensure a better consistency with the model scale and minimize the impacts of possible horizontal correlations on the observation error. A thinning at 0.5° spatial resolution, which consists in keeping 1

observation within a 0.5° grid box, is applied to both MODIS and PMAp. To reduce the large amount of VIIRS observations at 0.750 km, superobbing (Janjić et al., 2018) at the TL511 model spatial resolution, which consists in a spatial averaging of the observations at the model resolution, was applied to the VIIRS data.

A variational bias correction scheme (Dee et al, 2005; Benedetti et al., 2009), where biases are estimated by including bias parameters in the control vector, is applied to observations during the 4D-VAR minimization step. The bias correction is

continuously adjusted to optimise the consistency with all information used in the analysis. Currently, in the IFS, a bias correction can only be applied to an instrument and not a distinct instrument/satellite combination. Hence, in the CAMS operational configuration, the bias correction is applied only to PMAp and both TERRA/MODIS and AQUA/MODIS are used to anchor the bias correction, i.e. not bias corrected. However, given the differences observed between distinct satellite products in Garrigues et al., 2022, developments were made to the system to allow for different bias corrections to be applied to separate

instrument/satellite combinations. Therefore, when VIIRS is assimilated on top of MODIS and PMAp, we chose to apply the bias correction to MODIS, PMAp and SNPP/VIIRS. VIIRS AOD from NOAA20 is used as an anchor. The choice of anchoring NOAA20 and not SNPP was made based on the results from Garrigues et al., 2022 who reported a positive offset between VIIRS from SNPP and from NOAA20 over ocean due to a positive bias in the solar reflective bands of SNPP/VIIRS. The choice of applying the bias correction to both TERRA/MODIS and AQUA/MODIS is justified by the non-corrected

radiometric calibration degradation of TERRA/MODIS in the dark target algorithm and the aging of both instruments on board TERRA and AQUA with more frequent failure compared to VIIRS.

A fixed value of standard deviation of observation error (0.05 over ocean and 0.1 over land) has been used for MODIS since its implementation in 2009 (Benedetti et al., 2009). For PMAp and VIIRS, the pixel-level standard deviation of retrieval error provided with the product is used to represent the AOD observation error.

## 4 Methodology

### 4.1 Experiment design

Six-month simulations were conducted from 1/06/2020 to 30/11/2020. This period was selected because it encompasses major aerosol emission events such as the huge dust outbreak over the Mid-Atlantic in June (Yu et al., 2021) and the extreme and persistent fires in Siberia, North America and South America that occur from June



to September 2020. Figure 1 illustrates these events by showing the monthly departure of the CAMS reanalysis AOD from longer term averages, for the considered period.

Four experiments were designed to test distinct combinations of assimilated satellite AODs (Table 1). *PM* (PMAp and MODIS) corresponds to the CAMS operational configuration. The comparison of *PMV* (PM plus VIIRS) with *PM* gives indications on the impact of adding VIIRS on top of MODIS and PMAp for a future

operational configuration. The comparison of *V* (VIIRS-only) and M (MODIS-only) experiments informs on the impact of assimilating VIIRS in place of MODIS to understand how the differences in information content between MODIS and VIIRS AOD impact the analysis.

Table 1: Experiment characteristics (instruments used and applied bias correction; BC and A stand for bias

correction and anchor, respectively; NO stands for not used).

| | AOD Data assimilation | | | |
|---|---|---|---|---|
| Experiment name | PMAp | MODIS | VIIRS | |
| | | | NOAA20 | SNPP |
| PM | BC | A | NO | NO |
| PMV | BC | BC | A | BC |
| V | NO | NO | A | A |
| M | NO | A | NO | NO |

## 4.2 Evaluation methods

The first guess departure (FGD) quantifies the differences between the satellite observation (before bias correction) and its

model-simulated equivalent from the short-range forecast which was started from the previous analysis and is thus influenced by data assimilation. Positive values indicate that the observation is larger than the model and vice versa. The mean and the standard deviation (SD) of FGD represent the systematic and random, respectively, departures between the observation and the model. FGD is exploited in this work to i) characterize the spatial patterns of the departure between the observation and the model and ii) identify differences between products including products derived from the same

instrument but on-board distinct platforms (e.g., TERRA vs AQUA and SNPP vs NOAA20).

The overall differences between the experiments were assessed by comparing the daily average of the AOD analysis from each experiment. The daily analysis is computed as the mean value of the analysis obtained at 00h00, 06h00, 12h00,18h00 each day and then averaged over the JJA or SON period.



The accuracy of the AOD daily analysis was assessed against AERONET data (Holben et al., 1998; Giles et al, 2019) using
level 2.0 quality-assessed measurements which were available from 394 sites worldwide for the studied period. Model data
was interpolated to the observation sites and the observations themselves were averaged over 24-hour windows. The
performances are quantified through the Fractional Gross Error (FGE, Eq.1) and the Modified Normalized Mean Bias
(MNMB, Eq. 2). These metrics are insensitive to outliers in the distribution and their range is between 0 to 2 and −2 to 2,
respectively. They are computed at global scales considering all AERONET sites available, at regional scales (Europe, North
America, Southeast Asia and Africa), for 11 oceanic sites mainly influenced by sea salt aerosols and for 19 desert sites
mainly influenced by dust aerosols. The 24h temporal evolution of these metrics are provided for each experiment.

$$FGE = \frac{2}{n}\sum_{i=0}^{n}\left|\frac{AOD_i^m - AOD_i^o}{AOD_i^m + AOD_i^o}\right| \qquad \text{Equation (1)}$$

$$MNMB = \frac{2}{n}\sum_{i=0}^{n}\frac{AOD_i^m - AOD_i^o}{AOD_i^m + AOD_i^o} \qquad \text{Equation (2)}$$






in

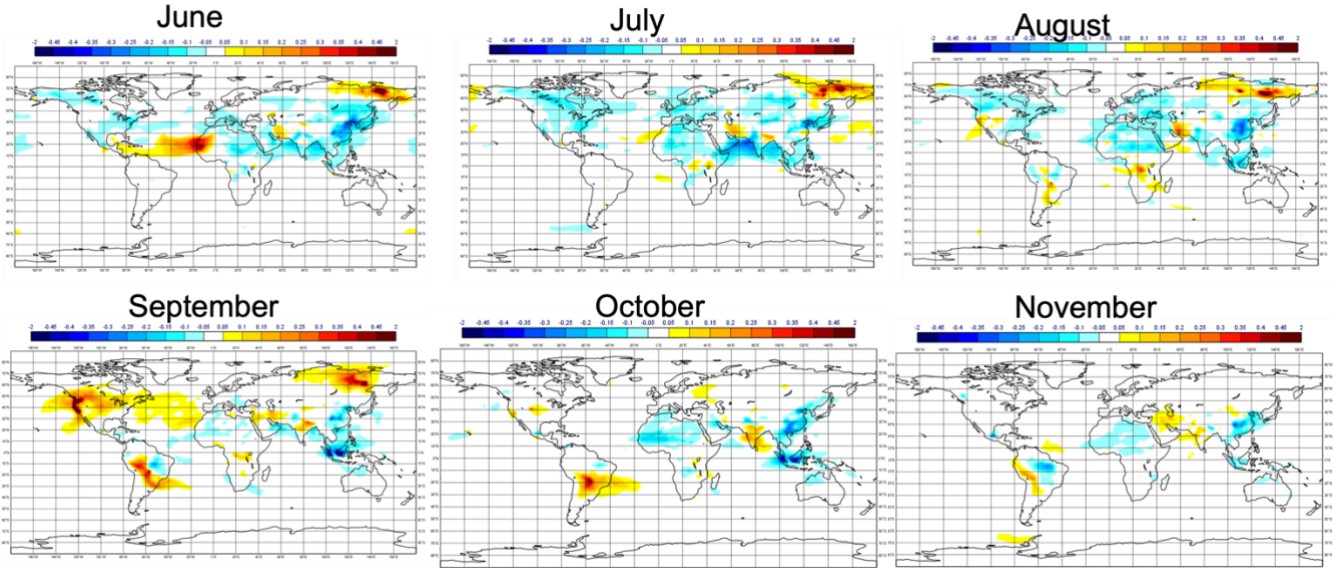

**Figure 1:** CAMS monthly AOD reanalysis anomalies in 2020 (computed as the difference between monthly average 2020-monthly average over the period 2003-2020)





## 4 Results

### 4.1 First guess departure spatial distributions.

Figures 2, 3, 4 and 5 show the ocean and land latitude transects of the mean and the SD of FGD for JJA and SON periods computed for the PMV experiment.

Over ocean, FGD of MODIS/TERRA is frequently positive and shows the highest values among the investigated AOD products. It is followed by FGD of MODIS/AQUA and VIIRS/S-NPP which have similar zonal distributions while FGD of VIIRS/NOAA20 is always negative and is the lowest among the investigated satellite AOD. Between 0° and 25°N, which includes the dust outbreak over the Mid-Atlantic, MODIS/TERRA, MODIS/AQUA and VIIRS/S-NPP show similar positive mean FGD. The lower mean FGD values of VIIRS/NOAA20 compared to S-NPP indicates a negative AOD offset between the two satellite products over ocean. Figure 4 shows similar SD of FGD between products in the Southern hemisphere while SD of FGD substantially increases for VIIRS in the Northern hemisphere (10°N to 60 °N for JJA and 30°N to 50 °N for SON) indicating larger random errors between the model and VIIRS AOD in this region. Overall, the products show small differences in the spatial distribution of their mean FGD between JJA and SON periods. However, the SD of FGD of VIIRS displays contrasted spatial distribution in the Northern hemisphere between the JJA and SON periods with overall higher values in JJA. Over land, the mean FGD of MODIS is frequently negative while that of VIIRS is frequently positive (Fig. 3). Both MODIS and VIIRS display similar negative mean FGD values over dust source regions (negative peak around 15°N in Fig.3a and Fig3.b). The largest differences in mean FGD between VIIRS and MODIS are obtained over biomass burning regions where VIIRS reach high positive FGD values. The seasonal variation of mean FGD over land mirrors the seasonal variation in regional fires: in JJA the differences between VIIRS and MODIS FGD occur above 50°N which is related to Siberia and North America fires and from 0 to 15°S related to South America fires; for SON larger discrepancies between VIIRS and MODIS appear in the SH due to the Amazonian fires in September. The relative differences in mean FGD between instruments on board different satellites are smaller over land compared to ocean except between NOAA20 and SNPP over dust source regions (0 to 25°N in JJA, Fig 3.a). The FGD SD is larger for VIIRS than for MODIS. SD of FGD of both VIIRS and MODIS increases over biomass burning regions and dust source regions with a larger increase for VIIRS. This indicates larger random differences between the model and the observations for dust and biomass burning aerosol types.

### 4.2 Analysis spatial distribution.

Figure 6 displays the maps of the differences in daily analysis between i) V and M experiments to see the impact of replacing MODIS by VIIRS and ii) PMV and PM experiments to see the impact of assimilating VIIRS on top of MODIS. Figures 7 and 8 compare the latitude transects over ocean and land, respectively, of the daily analysis AOD produced from those experiments. Overall, PM and M experiments show very similar analysis compared to the differences between the other experiments.

Over ocean, the assimilation of VIIRS decreases the analysis compared to the MODIS-based assimilation experiments. While the reduction is stronger when only VIIRS is assimilated (V experiment), the differences between PMV and V analysis are small compared to the differences between VIIRS-based assimilation experiments with those which do not assimilate VIIRS.





Over land, the assimilation of VIIRS leads to an increase of the analysis over biomass burning regions (e.g. North America, and Siberia in JJA and South America and Central and South Africa in SON) and a decrease over dust source regions (North Africa and Arabian Peninsula in JJA). The differences between V and M experiments are of similar magnitude compared to
the differences between PMV and PM experiments except over dust source regions where the decrease in analysis is stronger between V and M experiments.

## 4.3 Evaluation against AERONET measurements

Figure 9.a shows that the assimilation of VIIRS (PMV or V experiments) leads to an overall reduction of the AOD analysis
MNMB, evaluated against 394 AERONET stations at global scale, compared to the experiments based on the assimilation of MODIS (PM and M experiments). Bias reduction is relatively small in June and July, it substantially increases in a limited period of time in August (related to North America fires) and then it keeps steady values from mid-September to November. The assimilation of VIIRS also slightly reduces FGE but the reduction is relatively lower compared to the bias reduction. Figures 10 and 11 show regional evaluations of the AOD mean daily analysis. Figures 10.a and 10.b highlight the strong
reduction in AOD bias over desert and oceanic sites, respectively. At continental scale, the largest bias reduction occurs in Europe from September onwards (Fig. 11.a). Mixed results are observed over North America (Fig. 11b): the assimilation of VIIRS has very small impacts on the AOD analysis bias from June to July 2020; a large bias reduction is observed for the PMV experiment and negative biases for the V experiment in August 2020; the bias is then slightly reduced from October to November. For Africa (Fig. 11d), PMV and V experiments have smaller MNMB values than PM and M experiments and they
show negative biases in September and October. For South-East-Asia (Fig. 11c), all the experiments show a decrease of the bias in mid-August from 0.2-0.4 to reach 0-0.2 values in October. The assimilation of VIIRS has a small impact on MNMB from June to October and then leads to a slight bias reduction for the rest of the period. Results over South America are not shown since very little differences in AOD analysis between experiments were reported.







**a)**

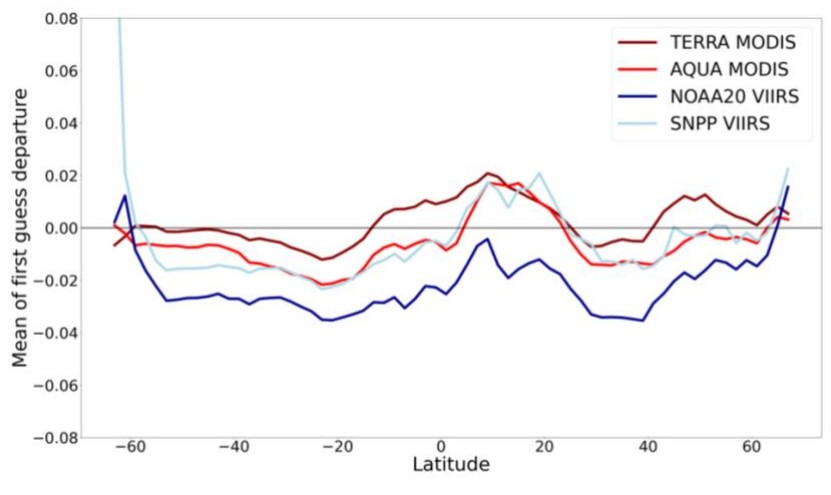

**b)**

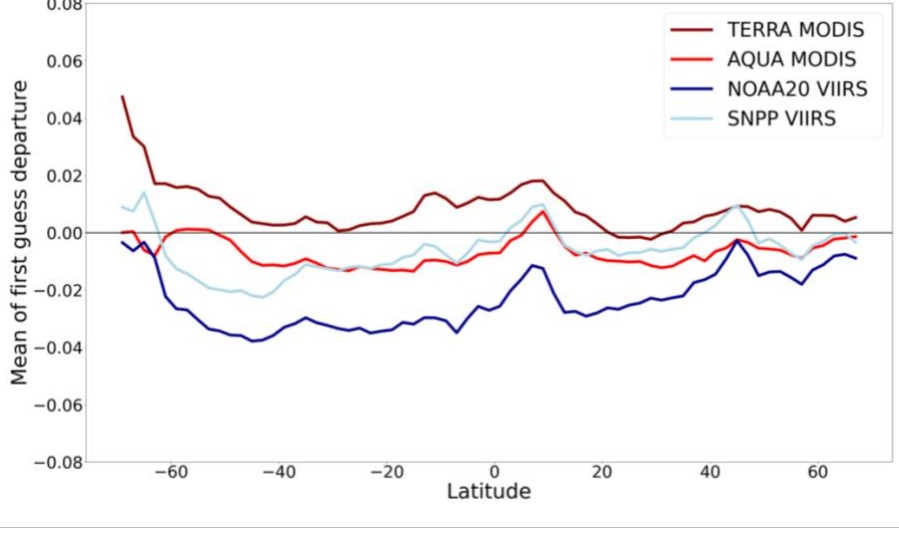


**Figure 2: Latitude cross-section of mean FGD, for the JJA (a) and SON (b) periods over ocean**





**a)**


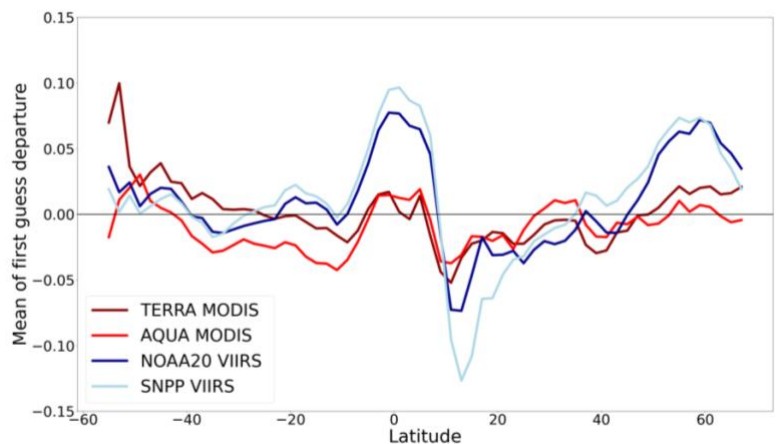

**b)**

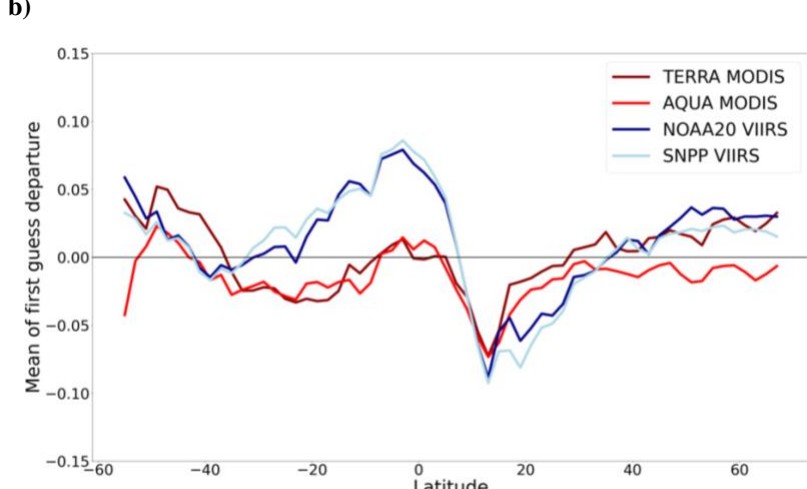

**Figure 3: Latitude cross-section of mean FGD, for the JJA (a) and SON (b) periods over land**







a)

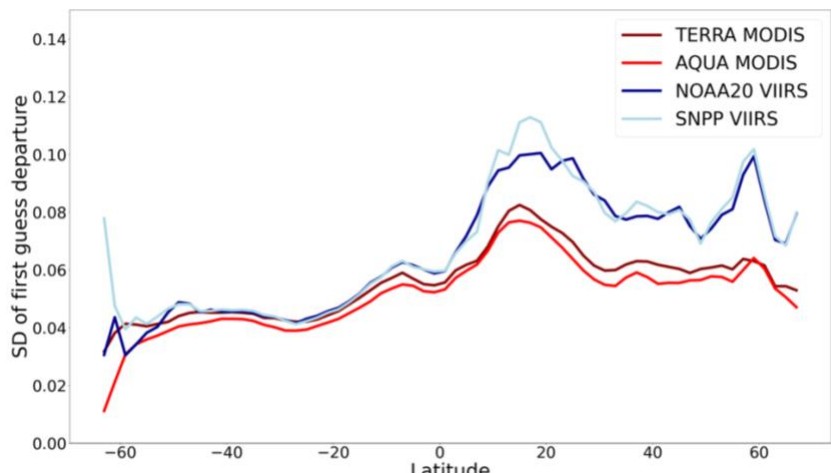

b)

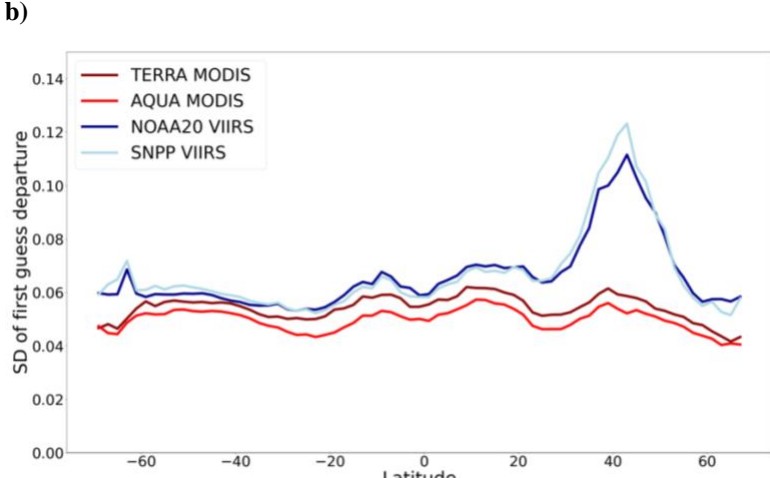


**Figure 4: Latitude cross-section of FGD SD, for the JJA (a) and SON (b) periods over ocean**






a)

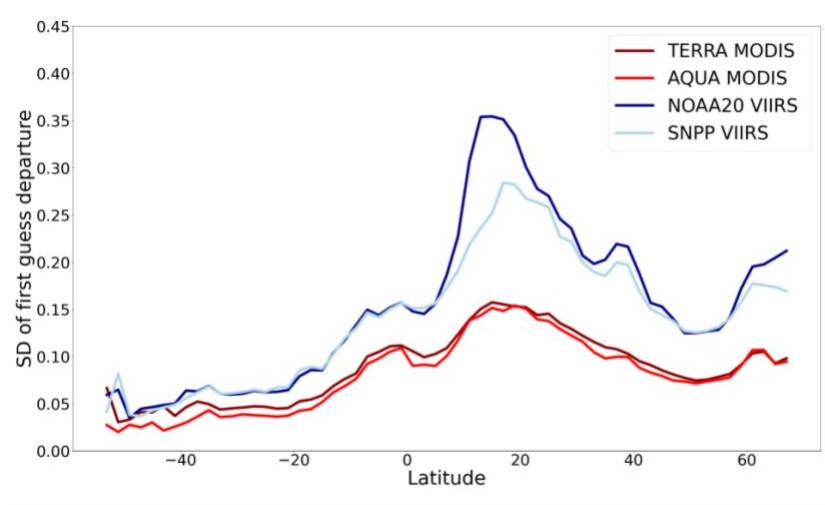

b)

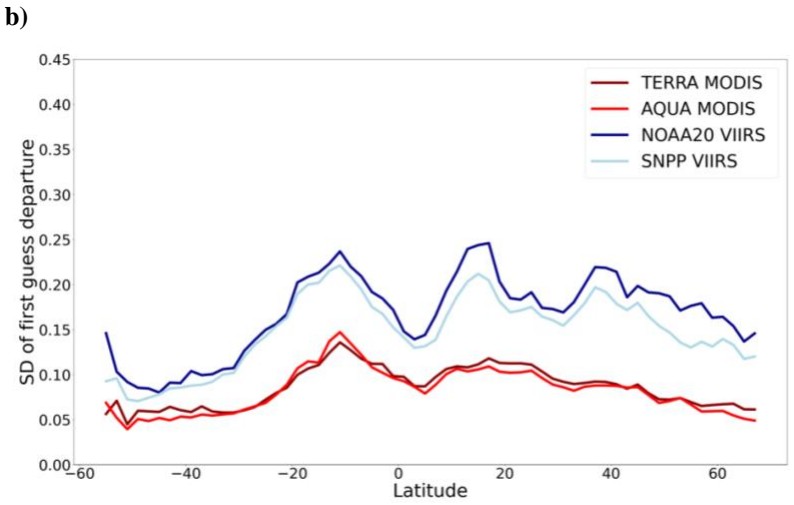

**Figure 5: Latitude cross-section of FGD SD, for the JJA (a) and SON (b) periods over land**



**a)**

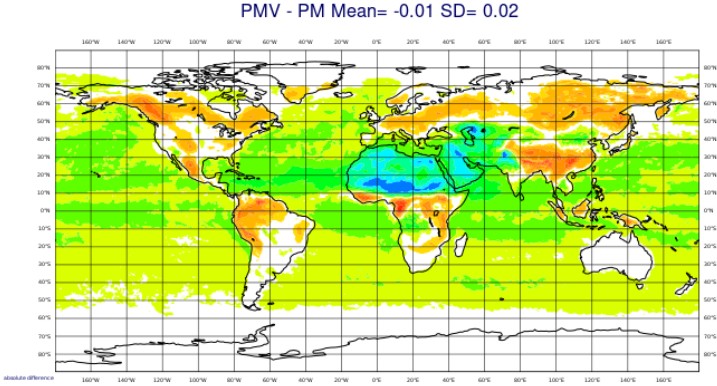

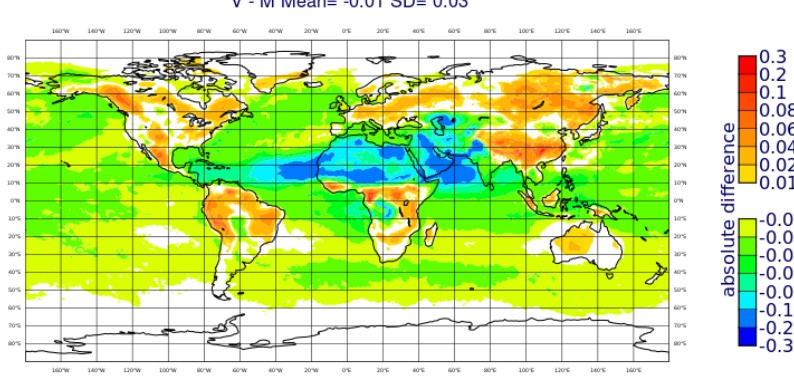

**b)**





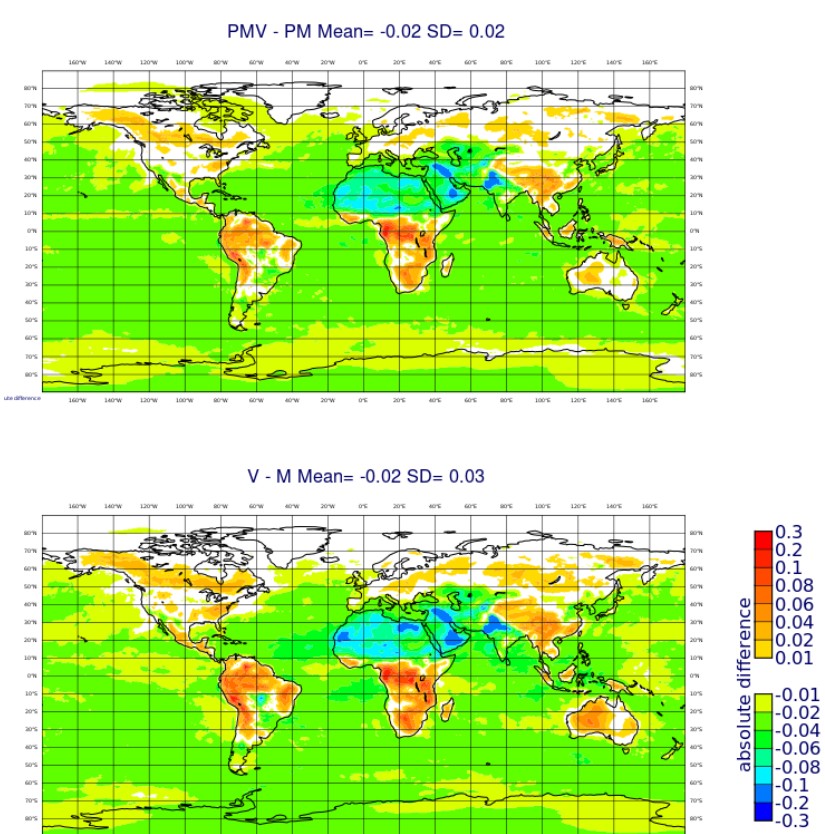

**Figure 6: Difference in mean JJA (a) and SON (b) AOD daily analysis between i) PMV and PM and ii) M and V experiments.**





**a)**

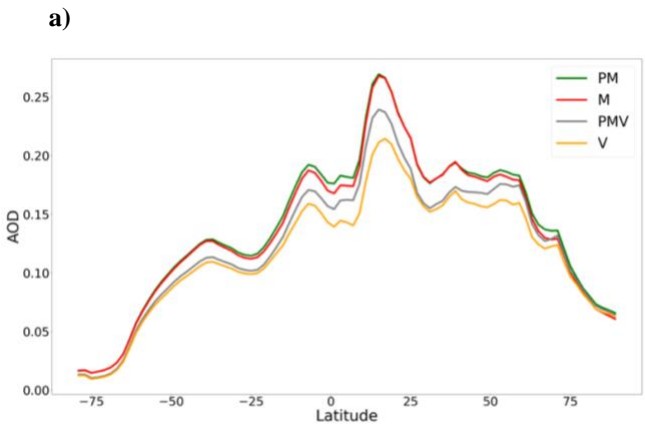


**b)**

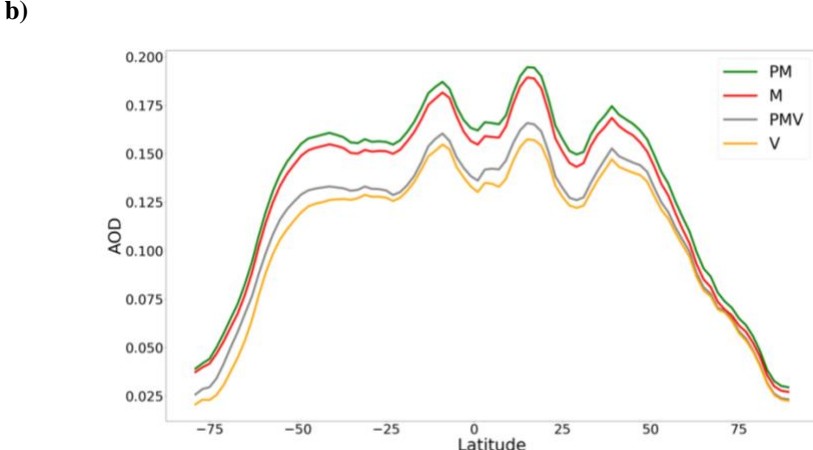

**Figure 7: Latitude transects of mean JJA (a) and SON (b) analysis over ocean**




a)

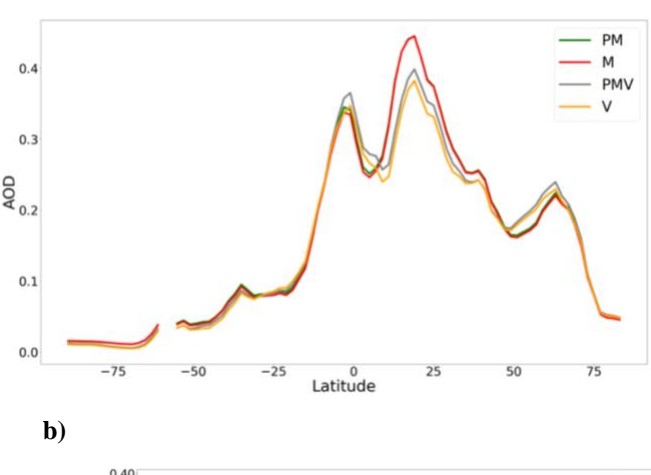

b)

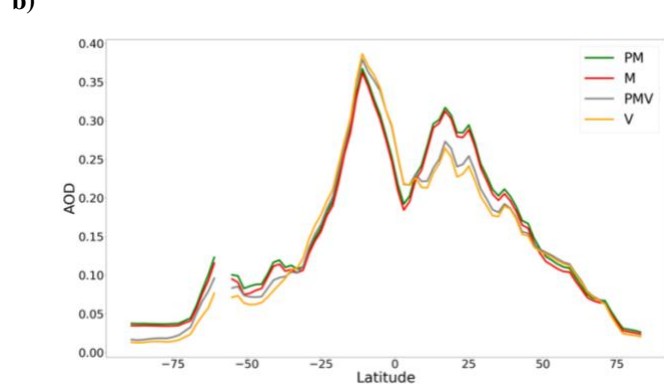


**Figure 8: Latitude transects of mean JJA (a) and SON (b) analysis over land.**




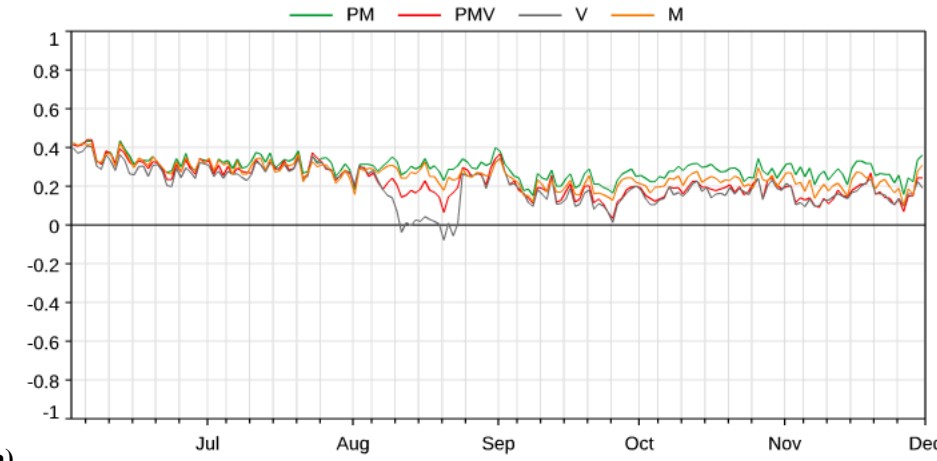

a)

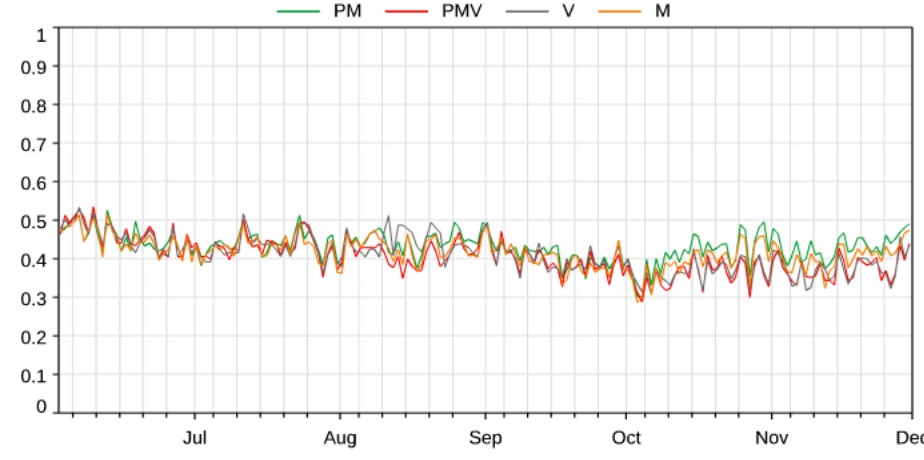

b)


**Figure 9: Temporal evolution of MNMB (a) and FGE (b) of the mean daily analysis of AOD for the PM, PMV, V and M experiments, evaluated against level-2 AERONET AOD measurements at 500nm from 394 sites at global scale.**


a)



a)

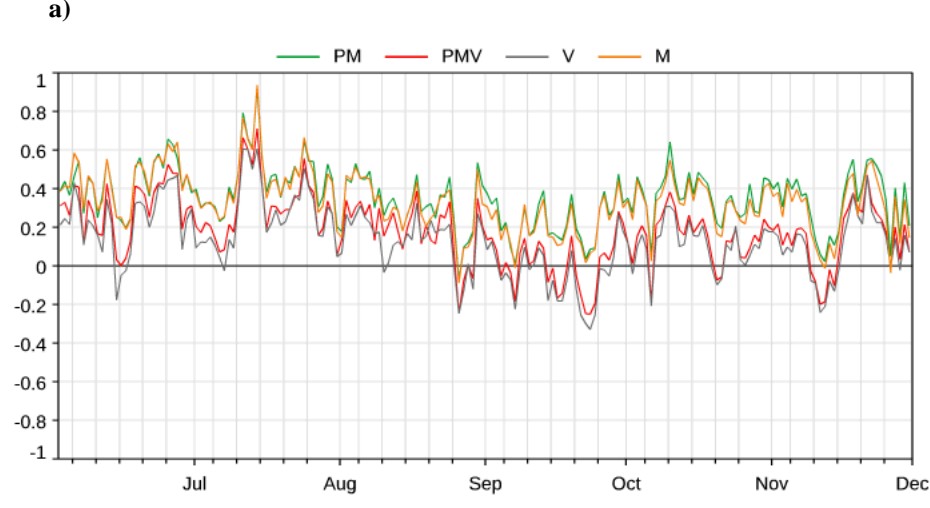


b)

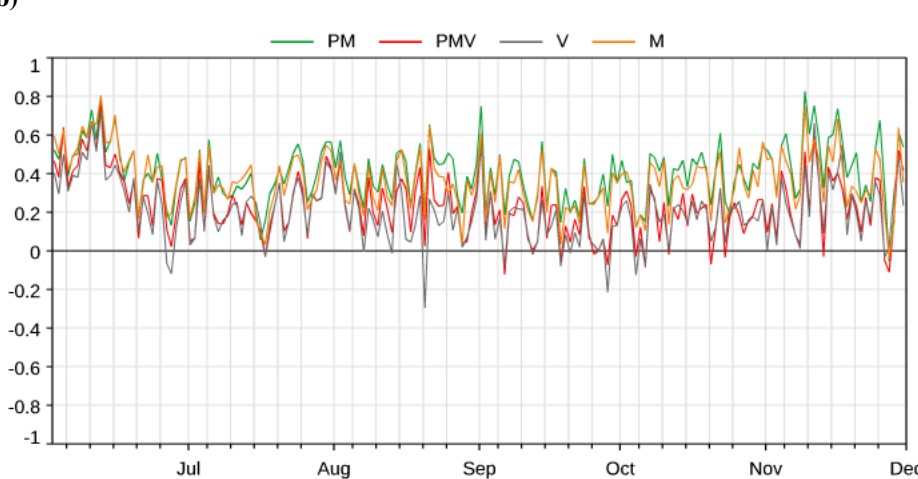

**Figure 10: Temporal evolution of MNMB of the mean daily analysis of AOD for the PM, PMV, V and M experiments, evaluated against level-2 AERONET AOD measurements at 500nm from 18 desert sites (a) and 11 oceanic sites (b).**



a)



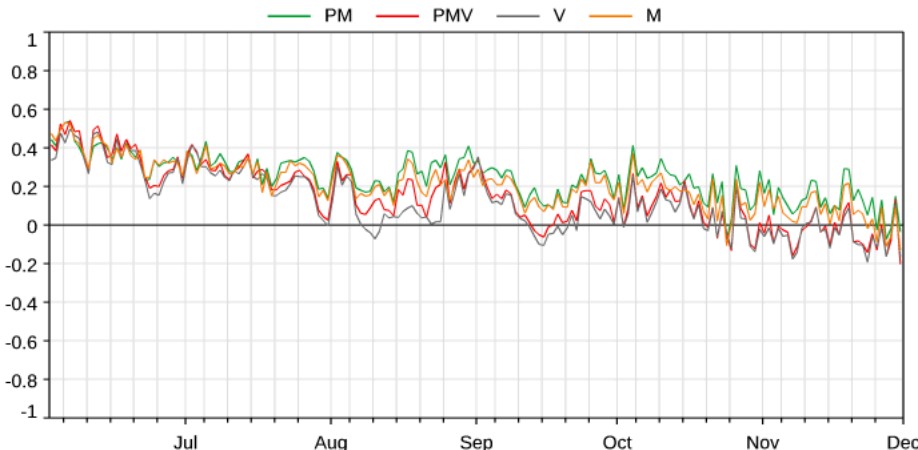


**b)**

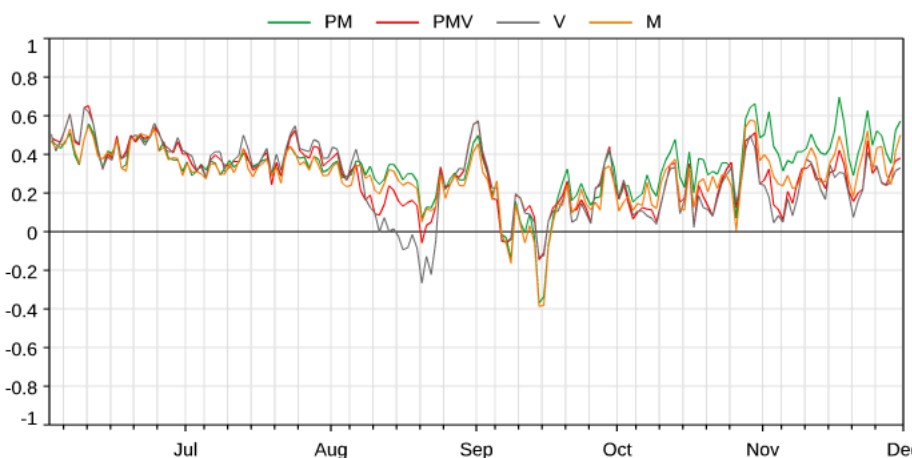



**c)**



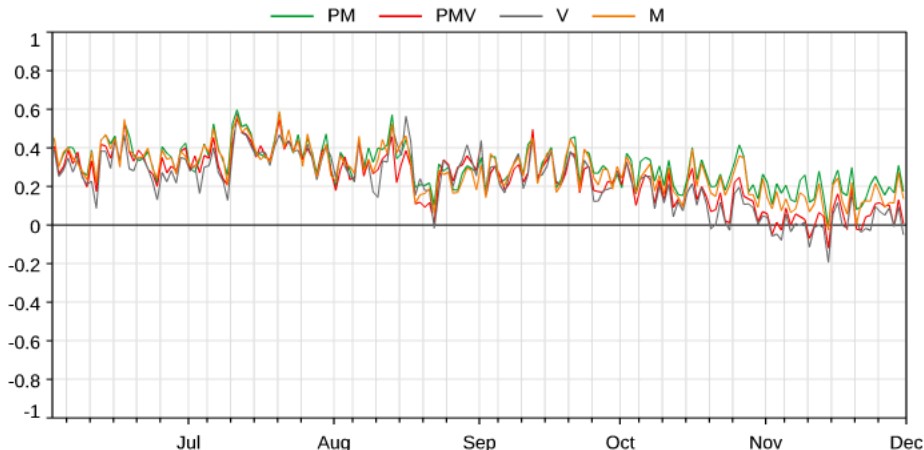

**d)**

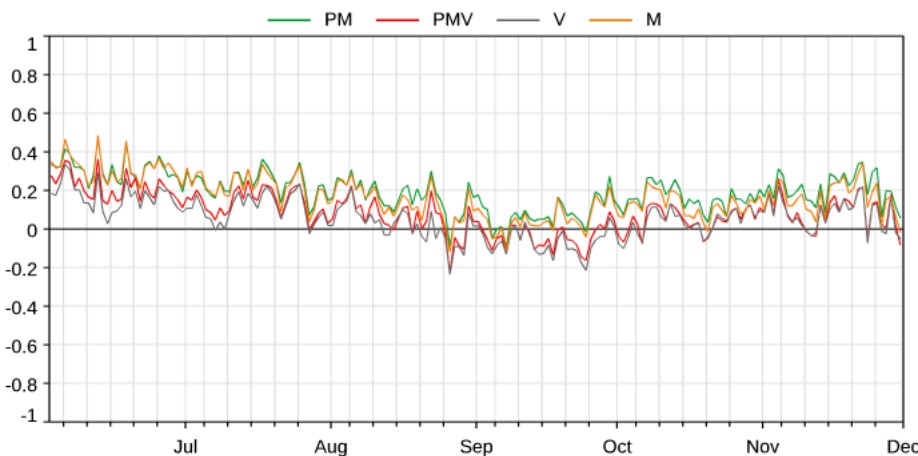

**Figure 11: Temporal evolution of MNMB of the mean daily analysis of AOD for the PM, PMV, V and M experiments, evaluated against level-2 AERONET AOD measurements at 500nm from 106 European sites (a), 130 North America**
**sites (b), 80 South-East Asia sites (c) and 53 Africa sites (d).**




## 5 Discussion

Compared to MODIS, the VIIRS AOD product represents a larger number of observations. Besides, because VIIRS algorithm is applying dynamically varying aerosol models during the retrieval, it potentially carries more information than the MODIS product which relies on prescribed and fixed in space and time aerosol models. We discuss here the results of this work to identify the expected changes in the CAMS aerosol analysis triggered by the assimilation of VIIRS.

### 5.1 Overall changes in the DA system

The implementation of VIIRS AOD observations in the CAMS DA system introduced significant changes affecting the AOD analysis when VIIRS is assimilated on top of MODIS. First, the choice of VIIRS/NOAA20 as an anchor while the bias correction is applied to MODIS (from both AQUA and TERRA) and VIIRS/SNPP plays an important role. This strategy was justified by 1) the known positive offset of MODIS/TERRA and VIIRS/SNPP reported over ocean by Garrigues et al., 2022 and 2) the frequent underestimation of MODIS AOD estimates over bright land surfaces as highlighted by Tao et al., 2017.

Second, the larger number of VIIRS cloud-free observations used in the DA, which is related to 1) the higher spatial resolution and the larger swath of the VIIRS instrument compared to MODIS and 2) the use of internal tests in the VIIRS retrieval algorithm to better distinguish between cloud and optically thick aerosol pixels, contribute to give more weight to VIIRS in the DA system. This is enhanced by the use of superobbing for VIIRS which exploit the information content of all best quality observations while the random thinning applied to MODIS may remove relevant observations with high information content.

This was illustrated in Garrigues et al., 2022 who reported that VIIRS resolved finer spatial details of smoke plume transport compared to MODIS which showed nosier spatial pattern.

### 5.2 Regional impacts

For background maritime aerosol, the assimilation of VIIRS decreases the analysis by 0.02 AOD on average that can have a

significant impact on the forecast given the low AOD value of maritime aerosols. The use of MODIS as an anchor in the operational configuration of CAMS (PM experiment) leads to large positive increments over ocean which are related to the known positive offset of MODIS/TERRA compared to MODIS/AQUA (Levy et al., 2018; Sogacheva et al., 2020). The assimilation of VIIRS and the use of VIIRS/NOAA20 as an anchor reduce the analysis increments providing more accurate AOD analysis for maritime sites as shown by Fig 11.b. Our results also show the positive offset between VIIRS/SNPP and

VIIRS/NOAA20 reported by Garrigues et al., 2022 for a different season in 2020. This offset is due to a known positive bias in the VIIRS/S-NPP top-of-atmosphere observed reflectances over ocean which translates directly in an AOD bias and thus justifies to bias-correct VIIRS/SNPP and anchor VIIRS/NOAA20.

Over land, the main changes triggered by the assimilation of VIIRS occur in dust and biomass burning regions as illustrated by the increase of the random departure of VIIRS AOD with the model (FGD SD) over these regions. For biomass burning

regions, the assimilation of VIIRS substantially increases the analysis increments which is related to the larger VIIRS AOD values than MODIS over fire areas. This confirms results from Garrigues et al., 2022 and is likely explained by 1) differences



in aerosol models between the VIIRS and MODIS retrieval algorithms and 2) the detection tests to identify heavy smoke aerosol in the VIIRS algorithm.

For dust, the assimilation of VIIRS strongly reduces the analysis over land dust source regions and dust outbreak over oceans.
This is particularly true in a latitude band between 10°N and 30°N where VIIRS produces smaller AOD than MODIS in JJA. However, these results contrast the findings from Garrigues et al., 2022 who show larger VIIRS AOD values over the Bodele depression in Africa, the Taklamakan desert and the Australian deserts during winter months in 2020.

The increase of increments from VIIRS experiments over China are likely related to complex interactions between natural and anthropogenic aerosols for which the differences in aerosol models between VIIRS and MODIS retrieval algorithm may
explain the larger VIIRS AOD than MODIS AOD (Garrigues et al., 2022).

Over Australia, the differences in analysis are larger between V and M experiments than PMV and PM experiments particularly in SON. VIIRS leads to larger analysis values than MODIS when it is assimilated in place of MODIS. The lower impact obtained in the PMV experiment suggests that the presence of MODIS mitigates the impact of VIIRS over this region.

**5.3 Seasonal dependency**

Over land, the seasonal variations are strongly driven by fire and dust seasonal variability: In JJA, VIIRS mainly influences North America and Siberia which both experience extreme fire events while in SON the large increments induced by VIIRS are concomitant with the South America and Central and South Africa fire seasons. The increments over China are more pronounced in JJA than in SON which may be related to both natural and anthropogenic seasonal variability.

**6 Conclusion**

The CAMS aerosol data assimilation system has been relying mainly on the assimilation of MODIS for more than 10 years. Given the aging of the MODIS instrument and consequently the possible disruption of the MODIS products, assimilating new aerosol observations has been a priority to ensure the continuity of the CAMS forecast performance. The objective of this work was to test the assimilation of the NOAA VIIRS AOD product from S-NPP and NOAA20 satellites in the IFS model and to
evaluate its impacts on the CAMS AOD analysis performances. In this work, experiments assimilating VIIRS on top of MODIS or in place of MODIS in IFS cycle 47R3 have been carried out from June 2021 to December 2022 which is representative of a large variability of aerosol events.

For ocean, VIIRS generally shows lower AOD values than the model and MODIS over maritime aerosol background. The assimilation of VIIRS and the use of VIIRS/NOAA20 as an anchor reduce the analysis compared to the MODIS-based
experiments which were producing too high increments due to the well-known positive offset of MODIS/TERRA over ocean. This study also confirms the positive bias of VIIRS AOD from S-NPP which justifies the choice of applying the bias correction to VIIRS/S-NPP and using VIIRS/NOAA20 as an anchor.




For land, VIIRS is larger than both MODIS and the model over biomass burning regions (e.g. Siberia, North America in JJA and South America, Central Africa in SON). This leads to an increase in the analysis when VIIRS is assimilated on top of MODIS or in place of MODIS. For regions with high dust load, including land and oceanic dust outbreak, VIIRS AOD is frequently smaller than MODIS and the model, which leads to a substantial decrease in AOD analysis. The assimilation of VIIRS reduces the increments over Sahara, Arabian Peninsula and some places in Asia in JJA. However, the impact of VIIRS on dust areas is seasonal-dependent as the changes in analysis are less important for SON 2021.

The comparison against AERONET shows an overall reduction of the bias in AOD analysis when VIIRS is assimilated either on top of MODIS or in place of MODIS. The largest bias reduction occurs in Europe followed by Africa and South-East Asia while more mixed results are obtained over North America. The assimilation of VIIRS substantially improve the analysis performances over desert and the bias reduction is significant over maritime aerosol sites given the low AOD of maritime aerosol background.

This work shows that the introduction of VIIRS in the CAMS DA system will strengthen the resilience of the CAMS DA system against the any disruption of the observing system which is an important aspect to ensure the continuity of the aerosol forecast performances. As VIIRS observations only include afternoon overpass, the next priority will be the implementation of morning overpass Low Earth Orbit (LEO) observations (e.g., Copernicus Sentinel-3 SLSTR) and geostationary products (e.g., GEMS, Copernicus Sentinel-4) to better sample the AOD diurnal cycle and mitigate the impacts of cloud cover that can have a strong diurnal cycle in some regions. Finally, future work should also focus on evaluating the observation error associated with each AOD retrieval which weight the contribution of each satellite product on the analysis increment and thus plays a key role in the implementation of a multi-satellite AOD DA system.




Code and data availability. Model code developed at ECMWF is the intellectual property of ECMWF and its member states, and therefore the IFS code is not publicly available. ECMWF member state weather services and their approved partners can get access granted to this code. Access to an open version of the IFS code (OpenIFS) that includes cycle CY47R3 IFS-AER may be obtained from ECMWF under an OpenIFS licence. More details can be found at https://confluence.ecmwf.int/display/OIFS/About+OpenIFS (last access:       )
(ECMWF, 2022). A software licensing agreement with ECMWF is required to access the OpenIFS source distribution: despite the name it is not provided under any form of opensource software license. License agreements are free, limited to non-commercial use, forbid any real-time forecasting, and must be signed by research or educational organizations. A detailed documentation of the IFS code is available from https://www.ecmwf.int/ en/publications/ifs-documentation (last access: 26 October 2022). The datasets used in this work and particularly the satellite AOD at the model grid resolution are available from https://apps.ecmwf.int/ research-experiments/expver/ (last
access: 26 October 2022)

Author contributions. SG carried out the simulations described in the paper and wrote the manuscript. RR set up the processing chain of each satellite AOD product at ECMWF to prepare their use in the ECMWF data system. MA and AI contributed to the analysis of the results and provided his expertise in data assimilation. SR contributed to the discussion by providing his expertise in aerosol modelling.
MP provided Figure 1 on AOD anomalies and helped with the result interpretation with respect to extreme fire event. The rest of the co-authors from the CAMS team provided valuable feedbacks on the use of multi-satellite AOD in CAMS and some develop and maintain the CAMS operational system.

Competing interests. The contact author has declared that none of the authors has any competing interests.

Acknowledgements. The Copernicus Atmosphere Monitoring Service (CAMS) is operated by the European Centre for Medium-Range Weather Forecasts on behalf of the European Commission as part of the Copernicus program (http://copernicus.eu, last access: 30 October 2022). The authors would like to thank the EUMETSAT, NASA, and NOAA space agencies for providing the satellite AOD products assimilated and monitored in CAMS.



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
