# Peer review of "Impact of assimilating NOAA VIIRS Aerosol Optical Depth (AOD) observations on global AOD analysis from the Copernicus Atmosphere Monitoring Service (CAMS)"

_EGUsphere, 2023_

## Author Response (AR1)

Dr Sebastien Garrigues

Copernicus Atmosphere Monitoring Service (CAMS)

European Centre for Medium-Range Weather Forecasts (ECMWF)

Shinfield Park, Reading, RG2 9AX, United Kingdom

E-mail: sebastien.garrigues@ecmwf.int

Tel: +44 (0) 118 949 9264

Reading, 08/06/2023,

Answer to final review of "Impact of assimilating NOAA VIIRS Aerosol Optical Depth (AOD) on AOD analysis from the Copernicus Atmosphere Monitoring Service (CAMS)".

Dear Editor,

Please find the revised version of the manuscript entitled "Impact of assimilating NOAA VIIRS Aerosol Optical Depth (AOD) on AOD analysis from the Copernicus Atmosphere Monitoring Service (CAMS)" along with the response to the reviewer's comments posted in ACPD.

We would like to thank all reviewers for their valuable feedbacks on the paper that helped to improve the quality of the paper.

We provide our response to each reviewer comment using red font. The changes in the revised version of the manuscript are highlighted in yellow.

**1- Response to reviewer 1**

*"The authors describe the assimilation of NOAA-VIIRS AOD in CAMS atmospheric model. This is an important study due to the drifting of MODIS Terra and Aqua satellites that will soon need to be replaced by VIIRS. The assimilation methodology is clearly defined, and the modeling results are clearly presented and explained. Overall, I recommend publication in ACP with two minor suggestions that might be taken into account by the authors: (1) I would encourage the authors to further extend the discussion on the sensitivity of the assimilation method towards specific aerosol types. This could be done for example by the evaluation of model performance at specific events when the dominant atmospheric composition is known (e.g. strong biomass burning or dust outbreaks). (2) I suggest that the discussion in Session 5 (especially 5.2 and 5.3) is based on bias improvement considerations (i.e. where and when the model performs better) rather than on the differences between the model runs."*

We agree with the two following suggested changes:

(1) *« I would encourage the authors to further extend the discussion on the sensitivity of the assimilation method towards specific aerosol types. This could be done for example by the evaluation of model performance at specific events when the dominant atmospheric composition is known (e.g. strong biomass burning or dust outbreaks). »*

Figure 10 provides the evaluation of the AOD analysis from each experiment against AERONET data for distinct groups of sites influenced by specific aerosol species. Fig. 10a shows the evaluation for desert sites which are mainly influenced by dust and Fig. 10b presents the evaluation for maritime sites which are mainly influenced by sea salt. In the revision, we added a similar regional plot for 47 West US sites which were influenced by the intense biomass burning events from mid-August to end of September 2020. All experiments have large negative and positive biases over this period compared to June-July and October-November. The AOD related to these extreme fires were poorly estimated by both MODIS and VIIRS, probably because of large cloud contamination and representativity uncertainties in the aerosol models. We added these new results in the revised version on p 9, line 225; p 12, line 308-312 and p 22, line 504.

>    (2) *« I suggest that the discussion in Session 5 (especially 5.2 and 5.3) is based on bias improvement considerations (i.e. where and when the model performs better) rather than on the differences between the model runs. »*

We modified the discussion in Section 5.2 to better emphasize where and when the assimilation of VIIRS leads to more accurate AOD analysis.

**2-  Response to reviewer 2**

"*In this manuscript, the authors present a study showing the impact of assimilating VIIRS AOD observations alone and jointly with MODIS AOD in CAMS. The paper is well written and is scientifically interesting, regarding that MODIS instruments are going to retire eventually. I recommend accepting it for publication.I have minor comments:*"

"*Abstract l19-20: I think the dates of your experiments need to be checked: The year is 2020 in your following text while in the conclusion it is 2021-2022.*"

We agree and we corrected the dates in the conclusion.

"*L23-24: "Over land, the assimilation of VIIRS induces a large increase in the analysis over biomass burning regions where VIIRS shows larger AOD than MODIS.": You may want to provide reasons on why VIIRS has larger AOD than MODIS?*"

VIIRS and MODIS exploit distinct aerosol models, which can explain part of the larger VIIRS AOD values. Tao et al., 2017 showed that the dust scattering properties are overestimated in the MODIS deep blue retrieval algorithm which results in a negative AOD bias over desert regions. Besides, VIIRS algorithm is applying dynamically varying aerosol models during the retrieval: it thus potentially carries more information than the MODIS product which relies on prescribed and fixed in space and time aerosol models. Another source of explanation is cloud contamination. The use of heavy smoke tests in the VIIRS algorithm leads to a better discrimination between aerosols and clouds and thus reduced cloud classification commission errors compared to MODIS. We better emphasized these explanations in the revised manuscript (p2, line 24) and (p28: lines 470-472 and lines 500-504)

"*L109-114: Any references about VIIRS NOAA algorithm should be provided.*"

The NOAA VIIRS AOD retrieval algorithm is described in  Laszlo, I. and Liu, H.: EPS Aerosol Optical Depth (AOD) Algorithm Theoretical Basis Document, NOAA-NESDISSTAR, Center for Satellite Applications and Research, https://www.star.nesdis.noaa.gov/jpss/documents/ATBD/ ATBD_EPS_Aerosol_AOD_v3.4.pdf (last access: 9 November 2022), 2020.

We added this reference in the product description Section 2 (line 94)

*"L155: You are assimilating total AOD at 550nm, however AOD distribution is non-gaussian. Do you take it into account in your assimilation system? Do you perform any quality control of your data using filters?"*

In the IFS assimilation system, both background and satellite observation error probability distribution functions are assumed to be Gaussian. We added this statement in line 140 page 6.

Distinct quality controls are applied to the observation data at different steps of the assimilation system:

- Prior to their use in IFS: the observation data are filtered by selecting only best quality retrievals according to the data provider recommendations. In CAMS, MODIS DT retrievals associated with a quality assessment (QA) equal to three over land and larger or equal to one over ocean are selected. DB retrievals associated with QA larger or equal to two are used to gap-fill DT over land. For PMAp and VIIRS, best quality retrievals are selected.

- Within IFS, a first filter consists in removing the observations that are far from the model first guess value. Then, a variational quality control, based on a Bayesian formalism, consists in reducing the analysis weight given to the observations which show large departure with the model first guess but still fall within an acceptable distance from the model (*Andersson and Janarvinen (1999)*.

  We introduced a paragraph dedicated to the quality control in the revised manuscript to better emphasize these aspects (page 7, line 170)

*"Section 3.3: You may want to revise your title to include VIIRS as well."*

We changed the title to include VIIRS as suggested.

Sincerely,

Sebastien Garrigues on behalf of co-authors